# Glass-Forming Ability and Corrosion Behavior of Ti-Based Amorphous Alloy Ti-Zr-Si-Fe

**DOI:** 10.3390/ma15207229

**Published:** 2022-10-17

**Authors:** Ling Bai, Ziyang Ding, Haiying Zhang, Chunxiang Cui

**Affiliations:** 1School of New Materials Engineering, Zhengzhou Technical College, Zhengzhou 450121, China; 2School of Materials Science and Engineering, Hebei University of Technology, Tianjin 300130, China

**Keywords:** glass-forming ability, corrosion behavior, amorphous alloy

## Abstract

Ti-based alloy Ti75Zr11Si9Fe5 (At %) and Ti66Zr11Si15Fe5Mo3 (At %) ribbons are fabricated by a single roller spun-melt technique, according to the three empirical rules. Both alloys are found to have a large, supercooled liquid region (ΔT_x_) before crystallization that reaches 80–90 K. The results show that both alloys possess excellent glass-forming abilities. The electrochemical measurement proves both amorphous alloys possess relatively high corrosion resistance in 3 mass% NaCl solution.

## 1. Introduction

Ti-based alloys have the advantages of high specific strength and strong corrosion resistance and are widely used as engineering materials [1,2,3]. At the same time, it has been found that amorphous alloys have superior mechanical properties, ductility and corrosion resistance compared to crystalline alloys [4,5,6,7]. Therefore, in recent decades, people have shown more and more interest in Ti-based amorphous alloys, and a lot of scientific research has been carried out [8,9]. Compared with crystalline Ti-based alloys, amorphous titanium alloys exhibit higher mechanical properties [10,11], low elastic modulus, excellent biocompatibility and high corrosion resistance. With the development of titanium-based amorphous alloys, Ti-rich bulk glassy alloys were found in rapidly solidified Ti-Cu-Al [12,13], Ti-Cu-Ni (Co) [14,15,16], Ti-Cu-Zr-Ni [17,18], Ti-Cu-Co-Al-Zr [19] and Ti-Cu-Ni-Sn (Si) [20,21] systems to date. Since 1988, many alloys with high glass-forming ability and capable of being formed into bulk metallic glasses (BMG) have been discovered more and more, for example, in multicomponent Mg-, La-, Zr-, Fe- and Pd-based metals in the alloy system. Later, Takeuchi and Inoue [22] classified BMGs into seven groups by studying the chemical species, internal atomic size differences and mixing heat of the constituent elements. This classification helps to grasp the various characteristics of BMG. In G-IV and G-VI, the Ti-based BMGs are generalized. However, the constituent elements are Ti-Cu-Ni and Ti-Zr-Cu systems, respectively. In this paper, Ti75Zr11Si9Fe5 (At %) and Ti66Zr11Si15Fe5Mo3 (At %) are fabricated successfully, whose constituent elements are Ti-Zr-Si. The main reason for choosing Ti75Zr11Si9Fe5 (At %) and Ti66Zr11Si15Fe5Mo3 (At %) in this research study is that the addition of alloying elements has a great influence on the phase composition and glass-forming ability of amorphous alloys. When selecting the added elements, it is necessary to consider not only the good glass-forming ability after addition but also the compatibility and harmfulness of the implants with the human body. For example, the metals Cu and Ni will cause cytotoxicity and neurotoxicity to the human body, while Al may cause Alzheimer’s and other symptoms. The novelty of this work includes three main aspects. In the first aspect, new ti-based amorphous alloys Ti75Zr11Si9Fe5 (At %) and Ti66Zr11Si15Fe5Mo3 (At %) are designed and developed for the first time. Secondly, Ti75Zr11Si9Fe5 (At %) and Ti66Zr11Si15Fe5Mo3 (At %) amorphous alloys have been prepared for the first time, which have good glass-forming ability. Thirdly, the corrosion resistance of the two new titanium-based alloys is significantly improved compared with pure titanium and Ti6Al4V alloys. The results show that both alloys possess excellent glass-forming ability. In addition, this paper also presents the thermal stability and corrosion behavior in NaCl solutions of both alloys.

## 2. Experimental Procedures

There are two main hypotheses in this study. Hypothesis 1, Ti75Zr11Si9Fe5 (At %) and Ti66Zr11Si15Fe5Mo3 (At %) have good glass-forming ability. Hypothesis 2, Ti75Zr11Si9Fe5 (At %) and Ti66Zr11Si15Fe5Mo3 (At %) amorphous alloys have good corrosion resistance. The alloy ribbons with a width of 5–8 mm and thickness of about 80 um were fabricated from the buttony Ti-based alloy ingots with nominal compositions of Ti75Zr11Si9Fe5 (At %) and Ti66Zr11Si15Fe5Mo3 (At %) by a single-roller spun-melt technique under an argon atmosphere. The phase composition and microstructure of the as-quenched alloy ribbons were examined by X-ray diffraction (XRD) using Cu K_α_ radiation and transmission electron microscopy (TEM), respectively. The thermal stability associated with glass transition temperature (T_g_) and crystallization temperature (T_x_) was investigated by differential scanning calorimetry (DSC) at a heating rate of 10 K/s. The corrosion behavior of both Ti75Zr11Si9Fe5 (At %) and Ti66Zr11Si15Fe5Mo3 (At %) was evaluated by electrochemical measurement. The electrolyte of 3% NaCl solution open to air was used at room temperature (about 298 k). The electrochemical measurements were conducted in a three-electrode cell using a saturated calomel (SCE) and graphite counter electrodes. The potentiodynamic cathodic and anodic polarization curves were measured separately at a potential sweep rate of 50 mV min^−1^. Hypothesis 1 can be supported by the XRD experiment and DSC experiment. Hypothesis 2 can be supported by electrochemical measurements.

## 3. Results and Discussion

### 3.1. Amorphous Structure of the Melt-Spun

Figure 1 below shows the melt-spun amorphous structure found in the patterns obtained from XRD experiments. From Figure 1, a broad peak can be clearly observed in the 2θ range of 31–48°, and binding experiments can analyze that this broad peak is produced by the amorphous phase. This indicates that both alloys designed have very good amorphous-forming ability. However, it can be seen from the figure that the glass-forming ability of the two alloys is not the same, and the alloy Ti66Zr11Si15Fe5Mo3 (At %) is better than that of Ti75Zr11Si9Fe5 (At %). The high glass-forming ability (GFA) of these two alloys by stabilizing the supercooled liquid region is mainly due to the formation of a new type of glassy structure; the structure has some special features, such as high-density packing, a new local atomic configuration and long-range homogeneity with attractive interactions [23,24,25].

In this paper, guided by the three empirical theories of Japanese scholar Inoue, combined with the theory of supercooled region before the crystallization of molten liquid in thermodynamics, as well as the most basic principle of a phase diagram, etc., the glass-forming ability of the two alloys was studied. Except for Pd-based alloys, most of the currently reported bulk glass alloys have those simple empirical composition rules. On the basis of a comprehensive study of important quantities such as alloying element additions, the different atomic size ratios and the interaction between constituent elements, Inoue outlined the three empirical rules [22]. First, the Ti75Zr11Si9Fe5 (At %) and Ti66Zr11Si15Fe5Mo3 (At %) alloys used in the experiment contain at least four types of elements, both of which exceed three. In the Ti66Zr11Si15Fe5Mo3 (At %) alloy, compared with the Ti75Zr11Si9Fe5 (At %) alloy, Mo is added, and the addition of this element makes the alloy more in line with the three empirical rules [22], which may be the reason the amorphous phase formation ability of the alloy is better than that of Ti75Zr11Si9Fe5 (At %). Second, the atomic size ratios between the main components of the two alloys both exceed 12% and are significantly different. The rearrangement of constituent elements of different atomic sizes may lead to higher packing densities. During the formation of titanium-based amorphous alloys, constituent elements of different atomic sizes are rearranged, and this process results in alloys with higher stacking densities. The atomic dimensions of the elements are derived from the atomic radii listed in the data book [26], which are calculated as half the distance between atoms in the crystalline state.

Table 1 lists the atomic radius of different elements. The atomic radii of both alloys exhibit the following rules: Si < Fe < Ti < Zr. The heat of mixing between the components of the alloy is negative. Since the crystallization of amorphous alloys requires a large amount of activation energy, the existence of atomic pairs with different negative mixing heats in amorphous alloys can greatly improve the thermal stability of supercooled liquids. In addition, in the amorphous alloy Ti66Zr11Si15Fe5Mo3 (At %), due to the addition of Mo, the degree of negative mixing heat between atomic pairs in the alloy is greatly deepened, such as the existence of atomic pairs such as Ti-Mo and Si-Mo, and this can increase the value of the negative heat of mixing in the alloy system. In addition, in the amorphous alloy Ti66Zr11Si15Fe5Mo3 (At %), due to the addition of Mo, the number of atomic pairs with negative heat of mixing, such as Ti-Mo and Si-Mo, can be effectively increased. Here, we express the heat of mixing between pairs of alloy atoms by the enthalpy of mixing between pairs of atoms containing equal atomic contents. The values of heat of mixing were quoted as the enthalpy of mixing ($\Delta H_{AB}^{mix}$) [27,28] of the binary liquid in an A–B systems at an equi-atomic composition.

Table 2 lists the values of $\Delta H_{AB}^{mix}$ (KJ/mol), for the atomic pairs between the two alloying elements were calculated according to the Miedema model. Generally, the mixing enthalpy of the solid alloy solution is affected by structural factors of the material itself, the chemical and elastic [29]. However, since, in a liquid (glassy) melt, there are no elastic and structural factors, in the research, the mixing enthalpy only considers chemical factors, and the mixing enthalpy can be determined based on the regular melt model [29,30], where $\Omega_{ij}$(=4$\Delta H_{AB}^{mix}$) is the regular melt interaction parameter between the *i*th and *j*th elements, *c_i_* is the atomic percentage of the *i*th component, and $\Delta H_{AB}^{mix}$ is the mixing enthalpy of binary liquid alloys.
$$\Delta H^{mix}=\sum_{i=1,i\neq j}^{n}\Omega_{ij}c_{i}c_{j}$$

The calibration value of $\Delta H_{AB}^{mix}$ is used as $\Delta H_{AB}^{mix}{}^{\left(cali\right)}=\Delta H_{AB}^{mix}-\Delta H^{trans}/2$ for containing one nontransition metal and $\Delta H_{AB}^{mix(cali)}=\left(\Delta H_{i}^{trans}+\Delta H_{j}^{trans}\right)/2$ for containing two nontransition metals. $\Delta H^{trans}$ are 100, 30, 180, 310, 17, 34 and 25 KJ mol^−1^ for containing H, B, C, N, Si, P and Ge [29], respectively. In summary, the enthalpies of mixing of the both alloys are −159.348 KJ/mol and −117.8748 KJ/mol, respectively. It is worth noting that the negative effect of the former alloy is greater than that of the latter alloy, which is part of the reason why the amorphous forming ability of Ti66Zr11Si15Fe5Mo3 (At %) is better than Ti75Zr11Si9Fe5 (At %). To sum up, the alloy system contains more than three alloys, the atomic size difference between the main components is greater than 12% and the large negative mixing enthalpy can lead to an alloy with a lower atomic dispersibility and glassy structure with a higher atomic stacking density. In general, the higher the stacking density of atoms, the better the thermal stability of the amorphous alloy and the greater the resistance of the supercooled liquid to the crystal transformation—that is, the higher the glass-forming ability of the amorphous alloy [31].

### 3.2. Microstructure Morphologies

The SEM microstructure of Ti75Zr11Si9Fe5 (At %) and Ti66Zr11Si15Fe5Mo3 (At %) is shown in Figure 2. It can be seen from Figure 2 that the SEM microstructures of the two alloys have relatively simple surface morphologies, showing a disordered glass structure. Compared with Ti75Zr11Si9Fe5 (At %), Ti66Zr11Si15Fe5Mo3 (At %) has smaller grains in the structure. The main reason is that the atomic content of metalloid element Si is increased in the alloy Ti66Zr11Si15Fe5Mo3 (At %), and there is a large atomic size difference between Si and the main metal element Ti and a large negative mixed heat; this greatly increases the topologically disordered arrangement of atoms or molecules in the three-dimensional space of the alloy system, which hinders the progress of crystallization.

The TEM morphology of Ti75Zr11Si9Fe5 (At %) and Ti66Zr11Si15Fe5Mo3 (At %) is shown in Figure 3. It can be seen from the figure that the grain boundaries of the matrix phases of the two alloys are blurred, and even no grain boundaries can be seen in some areas, which is very similar to the amorphous structure of metallic glass. At the same time, a large number of nanoparticles cause precipitation along the fuzzy grain boundaries of the alloy, especially at the intersection of multiple grain boundaries, and the particle size of the precipitates is significantly larger than the other grain boundaries. The main reason is that the second phase has a greater probability of nucleation in the high-energy region at the defect, and the grain boundaries, dislocations and vacancies belong to such high-energy regions, and, relatively speaking, the grain boundary is more conducive to the second phase: nucleation. In addition, a sufficient amount of elements in the alloy are beneficial to the formation of the β-phase structure, so that a sufficient amount of β-phase structure has been formed when the alloy is in the molten state. The lower phase can also appear in the higher energy grain boundary region in the form of a second phase.

### 3.3. Analysis of DSC

Figure 4 shows the DSC thermograms of amorphous alloys of both Ti75Zr11Si9Fe5 (At %) and Ti66Zr11Si15Fe5Mo3 (At %). From the figure, it can be found that there is an obvious and broad glass transition region. Due to the large atomic size difference and negative mixing heat among the multi-components of the bulk amorphous alloy, the nucleation and growth rate of the crystalline phase of the alloy are inhibited. As a result, a stable supercooled liquid phase region appears in the alloy before crystallization. It can be seen from the figure that there are differences in the supercooled liquid phase (ΔT_x_) of the two alloys before crystallization, which are about 80 K and 90 K, respectively, and this is not common in the Ti-based amorphous alloy system. The single exothermic peak is generated by the interaction of several crystal phases. The crystalline mode implies that the atomic rearrangements of the constituent elements on a long-range scale are necessary for the progress of the crystallization reaction. This inevitably leads to a delay in the crystallization reaction, resulting in a high thermal stability of the supercooled liquid. In such supercooled liquids, the topological and chemical short-range order will be greatly enhanced, and the structure will also change, which is conducive to the formation of high-density random packing structures, which are typically characterized by low atomic diffusivity. In general, the higher the stacking density of atoms, the better the thermal stability of the amorphous alloy and the greater the resistance of the supercooled liquid to the crystal transformation—that is, the higher the glass-forming ability of the amorphous alloy [31]. All these further improve that both Ti75Zr11Si9Fe5 (At %) and Ti66Zr11Si15Fe5Mo3 (At %) alloy ribbons with a large, supercooled liquid region possesses a high glass-forming ability and can be consolidated into a bulk form by taking advantage of the large viscous flow in the supercooled liquid region.

### 3.4. The Potentiodynamic Polarization Curve

In order to gain a deeper and more comprehensive understanding of the corrosion behavior of these two amorphous alloys, electrochemical measurement was performed in 3 mass% NaCl solution. Before the start of the corrosion test, the alloy specimens were mechanically polished in cyclohexane with silicon carbide paper up to grit 2000, degreased in acetone, washed in distilled water, dried in air and further exposed to air for 24 h for good reproducibility. Figure 5 shows the potentiodynamic polarization curve of both Ti75Zr11Si9Fe5 (At %) and Ti66Zr11Si15Fe5Mo3 (At %) amorphous alloys in 3 mass% NaCl solution open to air at 298 K. As seen in Figure 5, pure Ti and Ti6Al4V alloys are passivated with different current densities. However, both amorphous alloys kept steady current densities at about 0 A.m^−2^, which suggests that both were hardly eroded by the 3 mass% NaCl solution. Based on the data, it is concluded that the both amorphous alloys relatively possess excellent corrosion resistance compared with pure Ti and Ti6Al4V alloys in the Cl^-^-containing solutions or brine. Ti and Zr are known as strong passive valve metals in aggressive environments, and the enrichment of Ti and Zr in the surface film was found in Ti- and Zr-containing Ni-based glassy alloys [31]. This indicates that the improved corrosion resistance of the two amorphous alloys is mainly due to the formation of a protective film rich in Ti and Zr on the surface of the alloys in the NaCl solution. In addition, the improvement of corrosion resistance is also related to the structural and chemical uniformity of amorphous alloys.

## 4. Conclusions

(1)Ti−based alloys with nominal compositions of Ti75Zr11Si9Fe5 (At %) and Ti66Zr11Si15Fe5Mo3 (At %) have excellent amorphous-forming ability, which both satisfies the three empirical rules.(2)Both 75Zr11Si9Fe5 (At %) and Ti66Zr11Si15Fe5Mo3 (At %) amorphous alloys have a large, supercooled liquid region before crystallization, which is important to the amorphous-forming ability of the alloys.(3)The addition of Si and Mo improve the amorphous-forming ability of Ti66Zr11Si15Fe5Mo3 (At %) alloy.(4)Both 75Zr11Si9Fe5 (At %) and Ti66Zr11Si15Fe5Mo3 (At %) possess relatively excellent corrosion resistance, compared with pure Ti and Ti6Al4V alloys.

In this study, in order to ensure the accuracy of the two alloy compositions and the reliability of the results, it is suggested to test the composition of the two alloys before the experiment. Titanium-based alloys are widely used in biomedical fields because of their excellent biocompatibility and biological activity. Especially as a biological implant material, it plays an important role. The elastic modulus, excellent biocompatibility and excellent corrosion resistance make it develop rapidly. The results of this study about Ti75Zr11Si9Fe5 (At %) and Ti66Zr11Si15Fe5Mo3 (At %) will be beneficial to promote the development and application of titanium alloy implant materials.

## Figures and Tables

**Figure 1 materials-15-07229-f001:**
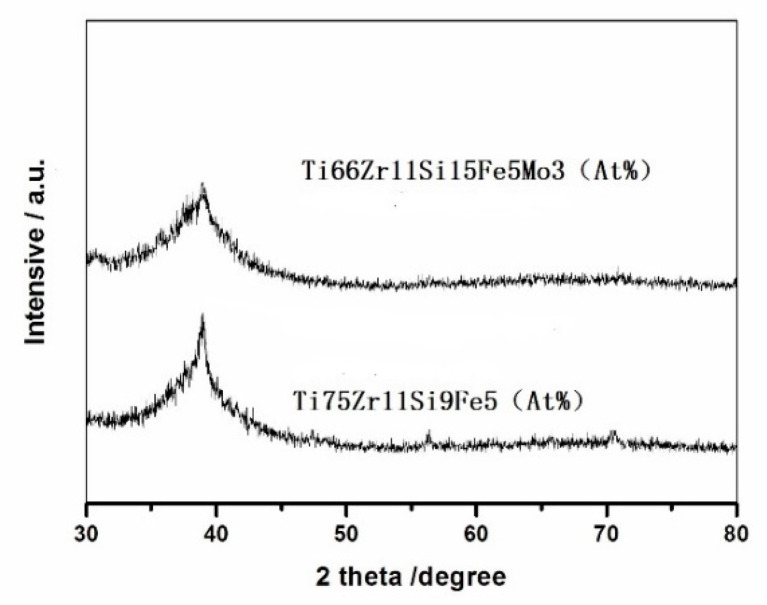
X-ray diffraction patterns of the as-quenched Ti66Zr11Si15Fe5Mo3 (At %) and Ti75Zr11Si9Fe5 (At %).

**Figure 2 materials-15-07229-f002:**
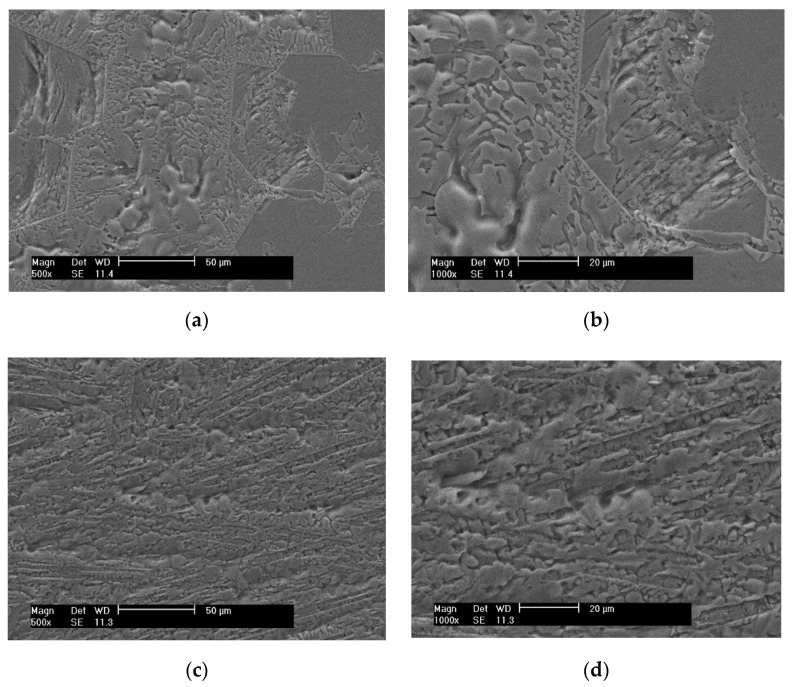
The SEM microstructure of Ti75Zr11Si9Fe5 (At %) and Ti66Zr11Si15Fe5Mo3 (At %): (**a**) Ti75Zr11Si9Fe5 (At %) (500×), (**b**) Ti75Zr11Si9Fe5 (At %) (1000×), (**c**) Ti66Zr11Si15Fe5Mo3 (At %) (500×) and (**d**) Ti66Zr11Si15Fe5Mo3 (At %) (1000×).

**Figure 3 materials-15-07229-f003:**
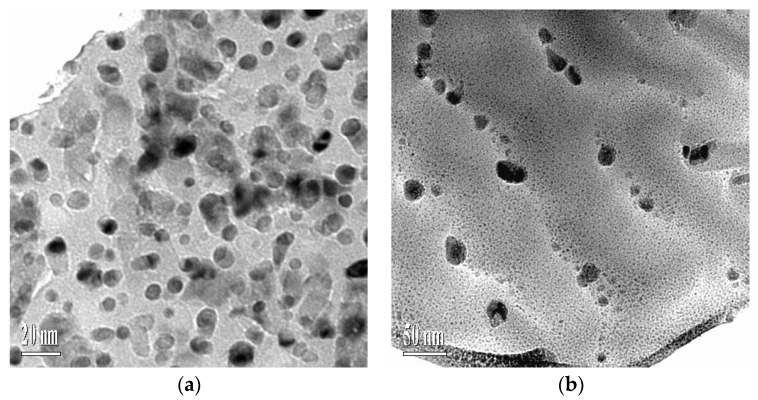
TEM morphology of Ti75Zr11Si9Fe5 (At %) and Ti66Zr11Si15Fe5Mo3 (At %): (**a**) Ti75Zr11Si9Fe5 (At %) and (**b**) Ti66Zr11Si15Fe5Mo3 (At %).

**Figure 4 materials-15-07229-f004:**
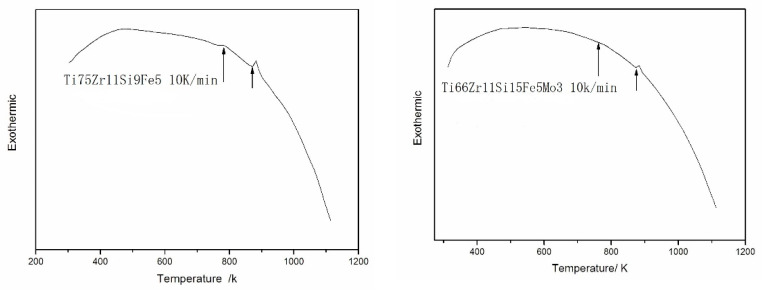
DSC thermograms of amorphous alloys of both Ti75Zr11Si9Fe5 (At %) and Ti66Zr11Si15Fe5Mo3 (At %).

**Figure 5 materials-15-07229-f005:**
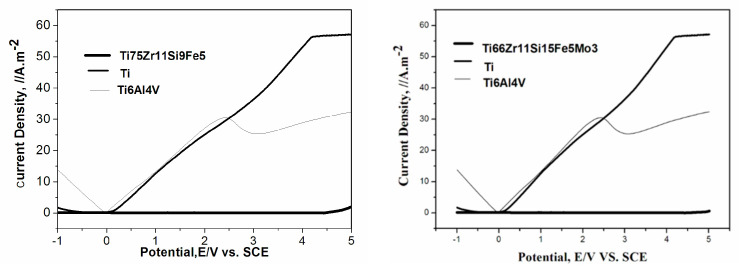
The potentiodynamic polarization curve of both Ti75Zr11Si9Fe5 (At %) and Ti66Zr11Si15Fe5Mo3 (At %) amorphous alloys in 3 mass% NaCl solution open to air at 298 K.

**Table 1 materials-15-07229-t001:** Atomic radius of the elements/nm.

Element	Ti	Zr	Fe	Si	Mo
Atomic radii	0.417	0.162	0.124	0.117	0.136

**Table 2 materials-15-07229-t002:** The values of $\Delta H_{AB}^{mix}$/KJ/mol calculated by Miedema’s model for atomic pairs between elements.

	Ti	Zr	Fe	Si	Mo
Ti	0	0	−17	−66	−4
Zr	0	0	−25	−84	−6
Fe	−17	−25	0	−35	−2
Si	−66	−84	−35	0	−35
Mo	−4	−6	−2	−35	0

## Data Availability

The data presented in this work are available on request from the corresponding authors.

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
