# Peer review of "Glass-Forming Ability and Corrosion Behavior of Ti-Based Amorphous Alloy Ti-Zr-Si-Fe"

_materials, 2022, doi:10.3390/ma15207229_

Round 1
Reviewer 1 Report
The article submitted for review entitled " Glass-forming ability and corrosion behavior of Ti-based amorphous alloy Ti-Zr-Si-Fe" raises the problem of Ti-based alloys which have the advantages of high specific strength and strong corrosion resistance, and are widely used as engineering materials. It is an interesting and comprehensive publication, in which the authors presented in a synthetic manner the results of their own research.
The findings obtained in the reviewed article on the basis of the experiment are a novelty.
The results are presented clearly and the conclusions are shortly formulated, highlighting the most important insights from the research.
Moreover i formulate a few remarks:
1) I suggest you increase the font inside the images (Fig.1, Fig. 4 – too small pictures).
2) The Conclusions are very poor and limited to few results. Please add more suggestions connected with achieved results or about perspectives of the future work.
Reviewer 2 Report
This study investigated the “Glass-forming ability and corrosion behavior of Ti-based amor- 2 phous alloy Ti-Zr-Si-Fe”. The current format needs a major revision and cannot be published at this stage;
Comments:
Ø The novelty of this work is not highlighted. This needs to be clearly explained. What was the hypothesis? How the results can support the hypothesis? Please explain why authors used these Ti75Zr11Si9Fe5 (At %) and Ti66Zr11Si15Fe5Mo3 (At %) in this research study?
Ø There should be a space between number and unit. For example, in section 2. Experimental procedures >>Line 49…. investigated by differential scanning calorimetry (DSC) at a heating rate of 10K/s……., etc. Please correct it through the paper
Ø English should be improved by a native speaker.
Round 2
Reviewer 2 Report
The authors done well and added most of the comments